# Humoral and Cellular Immune Response Elicited by the BNT162b2 COVID-19 Vaccine Booster in Elderly

**DOI:** 10.3390/ijms241813728

**Published:** 2023-09-06

**Authors:** Daniela Dalla Gasperina, Giovanni Veronesi, Carlo M. Castelletti, Stefania Varchetta, Sabrina Ottolini, Dalila Mele, Giuseppe Ferrari, Amruth K. B. Shaik, Fabrizio Celesti, Francesco Dentali, Roberto S. Accolla, Greta Forlani

**Affiliations:** 1Department of Medicine and Technological Innovation, University of Insubria, ASST Sette Laghi, 21100 Varese, Italy; d.dallagasperina@uninsubria.it; 2Research Centre in Epidemiology and Preventive Medicine (EPIMED), Department of Medicine and Surgery, University of Insubria, 21100 Varese, Italy; giovanni.veronesi@uninsubria.it; 3Molina’s Foundation, 21100 Varese, Italy; presidente@fondazionemolina.it (C.M.C.); giuseppe.ferrari@fondazionemolina.it (G.F.); 4Clinical Immunology-Infectious Diseases, Fondazione IRCCS Policlinico San Matteo, 27100 Pavia, Italy; s.varchetta@smatteo.pv.it; 5Department of Internal Medicine and Therapeutics, University of Pavia, 27100 Pavia, Italy; sabrina.ottolini01@universitadipavia.it; 6Microbiology and Molecular Virology Unit, Fondazione IRCCS Policlinico S. Matteo, 27100 Pavia, Italy; d.mele@smatteo.pv.it; 7Laboratory of General Pathology and Immunology “Giovanna Tosi”, Department of Medicine and Technological Innovation, University of Insubria, 21100 Varese, Italy; akbshaik@uninsubria.it (A.K.B.S.); roberto.accolla@uninsubria.it (R.S.A.); 8Center for Immuno-Oncology, Department of Medicine, Surgery and Neurosciences, University of Siena, 53100 Siena, Italy; fabrizio.celesti@unisi.it; 9Department of Medicine and Surgery, University of Insubria, ASST Sette Laghi, 21100 Varese, Italy; francesco.dentali@asst-settelaghi.it

**Keywords:** BNT162b2 mRNA COVID-19 vaccine, long-term care facilities elderly residents, neutralizing antibodies, Omicron BA.2, cellular immunity

## Abstract

Although the safety and efficacy of COVID-19 vaccines in older people are critical to their success, little is known about their immunogenicity among elderly residents of long-term care facilities (LTCFs). A single-center prospective cohort study was conducted: a total IgG antibody titer, neutralizing antibodies against Wild-type, Delta Plus, and Omicron BA.2 variants and T cell response, were measured eight months after the second dose of BNT162b2 vaccine (T0) and at least 15 days after the booster (T1). Forty-nine LTCF residents, with a median age of 84.8 ± 10.6 years, were enrolled. Previous COVID-19 infection was documented in 42.9% of the subjects one year before T0. At T1, the IgG titers increased up to 10-fold. This ratio was lower in the subjects with previous COVID-19 infection. At T1, IgG levels were similar in both groups. The neutralizing activity against Omicron BA.2 was significantly lower (65%) than that measured against Wild-type and Delta Plus (90%). A significant increase of T cell-specific immune response was observed after the booster. Frailty, older age, sex, cognitive impairment, and comorbidities did not affect antibody titers or T cell response. In the elderly sample analyzed, the BNT162b2 mRNA COVID-19 vaccine produced immunogenicity regardless of frailty.

## 1. Introduction

Elderly residents of long-term facilities (LTCFs) are at high risk of experiencing severe coronavirus disease 2019 (COVID-19), particularly if they have underlying chronic diseases [1,2,3].

To protect this population, in January 2021, the Italian plan for anti-SARS-CoV-2/COVID-19 vaccination established that elderly and extremely vulnerable LTCFs residents had priority in the vaccination schedule. 

The efficacy of the COVID-19 mRNA vaccine in inducing a strong systemic immune response against SARS-CoV-2 and in preventing severe disease has been well demonstrated [4,5]. However, LTCF elderly patients aged 70 to 80 years or older with frailty have not been represented in most vaccine development research, despite being earmarked as the earliest recipients in any national vaccination schedule [6,7,8,9].

A large epidemiological study on the effects of the BNT162b2 COVID-19 vaccine booster (third dose) during the B.1.617.2 (Delta) variant spread has demonstrated a significant, rapid, and consistent reduction in the COVID-19 burden among persons living in LTCFs [10,11]. Among this population, high protection against hospitalization and deaths from COVID-19, but modest protection against infection, was confirmed after the fourth dose during a substantial increase in the Omicron variant [12].

Therefore, even if the most recent studies in the elderly population also confirm the efficacy of mRNA COVID-19 vaccines against the variants of concern (VOCs), little is known about the humoral immune and T cells response. 

The significant loss of potentially protective antibodies has been shown within 6 months following the primary course vaccination with COVID-19 mRNA vaccines (2 doses) among elderly nursing home residents, particularly in those individuals without prior SARS-CoV-2 infection [13,14]. In addition, booster vaccination significantly increased vaccine-specific anti-spike, anti-RBD, and Omicron-specific neutralization activity above the pre-booster levels in nursing residents, both in those with and without prior SARS-CoV-2 infection [15,16].

This single-center prospective cohort study was conducted to evaluate both the humoral immune and T cells response elicited before and after the BNT162b2 booster (third dose) among LTCFs elderly residents who have undergone a primary vaccination cycle with the BNT162b2 COVID-19 vaccine for at least 6 months.

## 2. Results

### 2.1. Study Participants

Between October 2021 and January 2022, 49 vaccinated elderly residents from LTCFs with a median age of 84.8 ± 10.6 years were enrolled. Thirty-seven subjects (75%) were female. Previous COVID-19 infection (PI) was documented in 42.9% of the subjects at more than one year (median: 342 [331; 433]) before T0. Table 1 shows the clinical features of the participants. All the recruited subjects were not infected with SARS-CoV-2 in the period between T0 and T1. In the 6-month follow-up post-booster, nineteen subjects (38.7%) contracted a SARS-CoV-2 infection, all with mild symptoms. Indeed, none of them developed severe disease or required hospitalization, reinforcing the booster’s efficacy in preventing the severity of the disease. None developed significant adverse reactions after the vaccine doses. 

No significant differences in gender or age were observed between previous COVID-19-infected and not-infected individuals. The median time between T0 and T1 was 56 days; 42.9% of the subjects received flu vaccinations between T0 and T1.

### 2.2. BNT162b2-Vaccine Booster Increases Spike-Specific IgG Antiboiy Titer in Elderly

After the booster, the IgG antibody concentration increased in all the elderly subjects up to 13.6-fold (T0_GMC_: 361 IU/mL, T1_GMC_ 4910 IU/mL; Δ_T1-T0:_ 4549.2) (Table 2). This ratio was lower in the subjects with prior COVID-19 infections (up to 5.5-fold) because of the higher level of IgG antibodies at T0 compared with subjects not previously infected (T0_PI-GMC_: 1011 IU/mL and T0_GMC_: 166 IU/mL, respectively, *p* < 0.0001), suggesting that natural infection delays the decay of the antibody levels after the second dose of vaccination. At T1, IgG levels were similar in both groups (T1_PI-GMC_:4552 IU/mL and T1_GMC_: 4304 IU/mL, *p* = 0.64) (Figure 1 and Table 2).

### 2.3. High Neutralizing Activity against Wild-Type and Delta Plus, but Not against Omicron BA.2, Variants after BNT162b2 Vaccine Boost

Two doses of the BNT162b2 vaccine elicit high levels of protection from symptomatic disease, but this wanes over time. The BNT162b2 booster immunization can restore effectiveness to more than 90% of the general population, eliciting a strong systemic neutralizing activity. In order to verify the induction of specific neutralizing antibodies (Nab) in older people, we first analyzed the neutralization efficiency of the BNT162b2 booster vaccination against Wild-type SARS-CoV-2 (i.e., Wuhan) in 46 of the enrolled subjects. We found that 83% (38/46) of the subjects had neutralizing antibodies at T0 with an inhibitory activity (INH) mean of 81.6% (Figure 2). 

After the booster, neutralizing antibodies were found in all but one subject, and the neutralization efficiency slightly increased to 90.4%. (Figure 2). We next analyzed the neutralization efficiency of the BNT162b2 vaccine booster against Delta Plus and Omicron BA.2 VOCs as compared to T0. 

At T0, 74% (34/46) of the subjects presented Nab against the Delta Plus variant, with a mild reduction of INH means compared to Wild-type (INH: 76.88%). After the booster, Nab were found in 96% of the subjects (44/46), with INH mean values similar to Wild-type (INH: 90.4%). Of note, the individual lacking Nab against Wild-type after the booster also failed to develop antibodies against Delta Plus. Finally, only 41% (19/46) of the subjects presented Nab against Omicron BA.2 at T0, with a reduced INH (60%). The vaccine booster induced the production of Nab in 82.6% (38/46) of the subjects, with a slight increase in INH mean values (76.6%) (Figure 2). In conclusion, at T0, not all the subjects showed serum Nab against Delta Plus and much fewer against Omicron BA.2 VOCs. After the booster, the number of subjects with Nab and the neutralizing activity against Delta Plus was similar to that observed for Wild-type. Regarding Omicron BA.2, the number of subjects with Nab doubled, with a mild increase in INH. Those individuals without Nab against Wild-type did not develop Nab against either Delta Plus or Omicron BA.2 VOCs.

Interestingly, before the booster, there was a significant positive correlation between IgG titers and neutralizing antibodies against Wild-type and the two VOCs in the overall population and among SARS-CoV-2-negative subjects (Table 3). At variance, in subjects with prior COVID-19 infection, the correlation was positive and significant only for Omicron BA2. This can be explained by the fact that, among the previously exposed subjects, the percentage of Nab against Wild-type and Delta Plus was already substantially elevated, regardless of the serum IgG levels. For Omicron BA.2, neutralizing antibodies were found at IgG titers > 1000 or >10,000 IU/mL (Figure 3A, Table 3). At T1, the percentage of Nab response for Wild-type and Delta Plus was high regardless of the serum IgG titers. When observed at T0, after the booster, we found a positive correlation between serum IgG titers and percentage of Nab only for Omicron BA2.1, particularly in previously unexposed individuals (Figure 3B, Table 3).

Finally, at T0 and after the booster, no associations emerged between humoral response (serum IgG levels and %Nab) and age and gender. As far as the other comorbidities, IgG levels were lower for subjects with dementia (geometric mean: 3332 vs. 5133, *p*-value = 0.01) but without a similar finding as regards the percentage of Nab. Similar observations were found for chronic obstructive pulmonary disease (COPD) but with very few positive subjects (n = 7). No particular associations emerged for the other comorbidities (Table 4).

### 2.4. Induction of RBD-Specific T Cell Responses in Elderly after BNT162b2 Vaccination

T cell responses specific to the receptor-binding domain (RBD) were evaluated by ELISpot assay in PBMCs from 38 out of 49 vaccinated individuals after stimulation with RBD-15 mer overlapping peptides. A significant increase in T cell-specific immune response was observed in T1 as compared to T0 (Figure 4A). The median T cell responses were 74.5 (range: 1–540) and 21 (range: 0–312) IFN-γ SFU/10^6^ PBMC, respectively. 

In subjects with a prior COVID-19 infection, the median value of IFN-γ SFU/10^6^ PBMC was 105 (range: 1–540) at T1 and 37 (range: 0–312) at T0, while previously unexposed individuals exhibited a lower median value at both time points (74.5, range: 3–351 IFN-γ SFU/10^6^ PBMC at T1 and 20.5, range: 0–141 IFN-γ SFU/10^6^ PBMC at T0); however, these differences between previously unexposed and exposed subjects were not statistically significant at either T1 or T0. A significant increase in the response was observed instead at T1 compared to T0 among subjects without prior exposure to COVID-19 infection (Figure 4B).

Correlation analysis revealed a significant positive association between RBD-specific T cell responses and anti-spike antibodies exclusively at T0 (Figure 4C). Notably, this correlation was observed only in subjects with a previous natural infection (Figure 4D). However, no correlation was observed at T1, indicating a potential change in the relationship between T cell responses and antibody levels over time (Figure 4F,H).

## 3. Discussion

The impact of SARS-CoV-2 and COVID-19 disease susceptibility varies depending on an individual’s age and health status [17]. The elderly are more susceptible to SARS-CoV2 infections than younger individuals due to age and overall reduced function of the immune system [18,19,20]. Current vaccines have proven effective against severe disease caused by new SARS-CoV-2 variants, but to what extent they can protect the elderly population is still unclear, as the immunological correlates of protection in this population remains poorly studied [21,22,23,24,25]. Moreover, the appearance of new viral variants such as Delta (the predominant strain until mid-December 2021) and Omicron (emerged in November 2021 and rapidly spread throughout the world), as well as the reduction in the effectiveness of vaccination over time, has made the elderly population more susceptible to the infection. During our study, the local circulation of Delta variant was to a large extent superseded by Delta Plus and, particularly, by Omicron variants. Thus, we focused on analyzing neutralizing antibody activity against Delta Plus and Omicron BA.2, in addition to Wild-type strain, while T cell response was evaluated only against the original strain. We observed that the BNT162b2 booster induces a broad immune response with SARS-CoV-2-specific neutralizing antibodies and SARS-CoV-2-specific T cells in elderly people. Interestingly, in line with other reports, we found that the level of total IgG at 8 months from the first vaccination cycle significantly dropped in subjects unexposed to natural viral infection but remained high in subjects with prior COVID-19 infection diagnosed after the second dose of the vaccine [26]. Consistently, Demaret et al. observed that although COVID-19-naive older adults have a poor antibody response to the BNT162b2 mRNA vaccine compared with younger adults, the antibody response was greater in COVID-19-recovered older residents compared with unexposed COVID-19 subjects, reaching a level similar to that of young participants [27]. At variance, the booster dose significantly increased IgG titers, reaching similar values in all individuals regardless of prior infection [28,29]. At 8 months after the first vaccination cycle, Nab were found not only against the Wild-type strain but also against the Delta Plus VOC, with 83% and 74% of individuals showing similar levels of neutralizing activity. Interestingly, at this time point, the unexposed subjects developed Nab only at IgG concentrations above 100 IU/ml, suggesting that the level of IgG titer in the serum significantly affects the development of neutralizing antibodies and, thus, the protection against viral infection [30,31]. The neutralizing activity of the Ab increased after the booster, showing that the third shot of the BNT162b2 vaccine evokes a systemic immune response effective not only against the Wild-type strain (i.e., 90.4%) but also against the Delta Plus variant (i.e., 90.4%), as previously reported by others [29,32]. However, the scenario was completely different for the Omicron BA.2 variant. At T0, neutralizing antibodies were observed in the presence of high levels of IgG in the sera in all individuals, independent of previous COVID-19 natural infection. This might be explained by the fact that the recruited subjects were infected when the most predominant viral strains were the Wild-type and the Delta variant, and, therefore, natural neutralizing antibodies were poorly effective against the Omicron variant [33]. The booster doubled the percentage of the subjects with Nab against Omicron BA.2, with a slight increase in neutralizing activity (Nab, 82.6; INH, 76.6%). Accordingly, Muik et al. demonstrated that individuals between 20 and 72 years of age who received only two doses of the BNT162b2 mRNA vaccine had a low ability to neutralize the Omicron variant. In contrast, a third shot of BNT162b2 significantly improved antibody recognition of Omicron, suggesting that three doses of the mRNA vaccine BNT162b2 may protect against Omicron-mediated COVID-19 [34]. Among the unexposed subjects, we found a positive correlation between IgG levels and Nab, possibly indicating that, although the IgG titers are similar in SARS-CoV-2 unexposed and previously exposed subjects after the booster, natural infection might select B cell clones able to recognize and neutralize several viral epitopes conserved among the different mutated strains. Besides neutralizing antibodies representing the first layer of adaptive immunity against COVID-19, T cell responses play a crucial role as a second layer of defense in preventing severe COVID-19. BNT162b2 was demonstrated to induce a broad cellular immune response with poly-specific CD4^+^ and CD8^+^ T cells at least 9 weeks after the booster [7,35]. Even though emerging SARS-CoV-2 variants can elude recognition by neutralizing antibodies, especially when the level of antibodies declines over time, cell-mediated immune responses are much better sustained [36,37,38]. Here, we found that at T0, the T cell response specific to RBD was higher in previously exposed individuals compared with unexposed subjects (median: 37 [0–312]; median 20.5 [1–540]), respectively. These results are in agreement with previous reports suggesting that patients with prior COVID-19 infection had a better T cell response compared with unexposed subjects [27]. Of note, Hansen et al. have evaluated both humoral and cellular immune response in naturally infected older people and in uninfected older people 9 months after the first dose of the BNT162b2 vaccine. They observed that T cell responses persisted up to 12 months only in naturally infected older patients but waned in previously unexposed subjects. Accordingly, in our study, T cell response was low at T0 in vaccinated older people. Importantly, and similarly to our results, they observed that pre-infected older adults had more robust and durable antibody responses compared with unexposed individuals [39]. Cumulatively, these results emphasize that natural infection induces more robust and durable immune responses in the elderly, which is not achieved by two doses of the BNT162b2 vaccine. 

The booster pulsed cell-mediated immune response in all subjects. Interestingly, only in individuals with prior natural infection did we find a significant positive association between RBD-specific T cell responses and IgG levels exclusively at T0. Higher Ab titers and more specific T cells most likely correlate to the increased number of memory T cells generated by vaccination/natural infection and thus to the increased primed T cell-dependent stimulation of antibody-producing B cells. 

The correlation between T cell responses and anti-spike antibody titers was previously described by other groups, particularly in naturally infected pre-exposed individuals [40].

Consistently with our findings, Hurme et al. reported that, in working-age vaccinated individuals, the levels of S1-specific antibodies did not correlate with T cell responses, whereas, in the COVID-19 patients’ anti-S1 IgG antibody levels, a correlation trend with T cell responses was observed [35]. Consistently As, in the present study, we evaluated the T cell response only against the original strain of the virus, it will be crucial in the future studies to investigate the persistence of BNT162b2 vaccine-induced cellular immunity to the most prominent VOCs. Even though we clearly demonstrated that the booster dose significantly elicits both neutralizing and cellular immune responses in the elderly, as was observed for younger people, the weakened immune system of the elderly, characterized by steady decline of innate and adaptive immune responses, should be taken into consideration in future strategies of vaccine optimization for different age groups.

## 4. Materials and Methods

### 4.1. Study Design and Participants

We performed a single-center prospective study recruiting elderly residents of an LTCF in Northern Italy (Fondazione Molina, Varese, Italy), who underwent the BNT162b2 (Pfizer-BioNTench^®^) boosting vaccination by intramuscular injection (i.e., third dose) to evaluate the systemic humoral immune response elicited by the vaccine. 

The inclusion criteria were the age of 70 years or older, completion of a primary vaccination cycle with two doses of BNT162b2 COVID-19 vaccine within the past 6 months, negativity for SARS-CoV-2 infection as assessed by COVID-19 antigen rapid test at the time of recruitment, absence of symptoms of acute infection, and acceptance to receive the booster dose injection of the BNT162b2 vaccine. 

Exclusion criteria were documented SARS-CoV-2 infection in the last 6 months and ongoing therapy with glucocorticosteroid and/or immunosuppressant.

Patients who arrived at the residential facility and met the inclusion criteria were prospectively recruited; the recruitment period started on 15 October 2021 and lasted 3 months. Patients who could not be evaluated after the third-dose vaccination because they died or were hospitalized for an unrelated cause were excluded from the study. No eligible patients in the study declined to participate. The clinical protocol for sample and data collection was approved by the Institutional Ethics Committee. The study was conducted in accordance with the Declaration of Helsinki. 

Data were collected 8 months after the second shot of the vaccine (T0) and at least 14 days after the booster (T1). A 15 mL sample of whole blood was obtained from each participant at T0 and T1: 5 mL into a container with a gel separator for serum fraction and 10 mL into an EDTA tube for peripheral mononuclear cells (PBMCs) isolation. 

The following clinical data were recorded: age, sex, main comorbidities (diabetes, chronic obstructive pulmonary disease [COPD], cardiovascular disease [CVD], dementia, autoimmune diseases), exposure to SARS-CoV-2 infection before T0, between T0 and T1 and/or within 6 months after the booster shot, and flu vaccination prior to T0 or between T0 and T1.

### 4.2. Antibody Measurement

A commercial enzyme-linked immunosorbent assay (ELISA) specific for the S1 protein of SARS-CoV-2 was used to measure IgG Ab titers in serum samples, according to the manufacturer’s instructions (Anti-nCoV19 S1 IgG HS Immunospark, Pomezia, Italy). Serum samples were analyzed at 1:10, 1:100, and 1:1000 dilution at T0, and at 1:100, 1:1000, and 1:10,000 at T1. The results were expressed as UI/mL, and the lower threshold established by the manufacturer was 0.625 IU/mL for the IgG Ab, as previously described [41].

Furthermore, the presence of anti-RBD Nab in serum was assessed by competitive ELISA, following the manufacturer’s instructions (cPASS^TM^ SARS-CoV-2 Neutralization Antibody Detection Kit, GenScript, Piscataway, NJ, USA), as previously described [41]. Briefly, 10 µL of serum were diluted in 90 µL of sample dilution buffer. Positive and negative serum controls provided within the kit were used as reference for the serum Nab. The optical density (OD) average of the negative controls was used to calculate the percentage of inhibition according to the following formula: (1 − OD value of the sample/OD value of negative control) × 100%. A cut-off value of 30% was used to discriminate between the presence or absence of Nab, according to the manufacturer’s instructions. In the case of positivity, the percentage of inhibitory activity (i.e., INH) was also assessed.

Finally, to verify the efficacy of the immune response against the VOC, the serum samples at T0 and after the booster were also tested against the RBD of the Delta Plus variant (lineage B.1.617.2.1; K417N, L452R, T478K) and the Omicron variant B2.A (lineage B.1.1.529.2; G339D, S371F, S373P, S375F, T376A, D405N, R408S, K417N, N440K, S477N, T478K, E484A, Q493R, Q498R, N501Y, Y505H).

### 4.3. Peptide Pools and Antigens

Fifty-three 15-mer peptide pools overlapping by 11 amino acid residues, representative of the receptor-binding domain (RBD), were used at the concentration of 3 μg/mL per well (ChinaPeptides, Shanghai, China).

### 4.4. Ex Vivo Enzyme-Linked Immunospot Assay (ELISpot Assay)

PBMC were rested for 2 h in complete medium and seeded at 3.33 × 10^5^ cells/well in 96-well plates pre-coated with anti-IFN-γ (15 μg/mL; clone 1-D1K; Mabtech, Nacka Strand, Sweden). Test wells were supplemented with the 15-mers described above. Negative control wells lacked peptides, and positive control wells included PHA (5 μg/mL; Sigma-Aldrich, St. Louis, MO, USA). Tests were performed in duplicate. Samples were incubated for 24 h at 37 °C. Plates were then washed five times with PBS (Lonza, Basel, Switzerland) and incubated for 2 h at room temperature with biotinylated anti-IFN-γ (1 μg/mL). After five further washing steps, a 1:1000 dilution of alkaline phosphatase-conjugated streptavidin (Mabtech, Nacka Strand, Sweden) was added for 1 hr at room temperature. Plates were then washed a further five times and developed for 20 min with BCIP/NBT Substrate (Mabtech, Nacka Strand, Sweden). Spots were counted using an automated ELISpot Reader System (Autoimmun Diagnostika GmbH, Strasburg, Germany). Results were given as IFN-γ spot-forming units (SFU)/10^6^ PBMC after calculating the mean values from duplicate wells and subtracting spots from negative control. The positive cut-off was set at 10 IFN-γ SFU/10^6^ PBMC.

### 4.5. Statistical Analysis

Participant characteristics at T0 were summarized using standard descriptive statistics. To account for their skewed distributions, at each visit time, we calculated sample medians and interquartile range for serum IgG, in the overall sample and according to previous SARS-CoV-2 infection. To estimate time trends in serum IgG, we used repeated-measure regression models, with baseline value, time visit, previous SARS-CoV-2 infection status, and the interaction between time and infection status as independent variables. Again, we modelled log-transformed IgG values and reported the geometric mean concentrations (GMC) with 95% confidence intervals. Individuals were categorized according to the presence of Nab. We estimated the linear correlation between log-transformed serum IgG levels and VOC-specific Nab positivity at T0 and T1 using the Pearson’s correlation, in the overall sample and by previous SARS-CoV-2 infection. We tested whether the prevalence of Nab varied with virus variance (i.e., Wuhan Wild-type, Delta Plus, and Omicron BA.2 strains) using a regression model with repeated-measure (unstructured variance-covariance matrix) and including an interaction term between sample and variant (3 degrees of freedom, Wald chi-square test). Statistical analyses were carried out using the SAS Software (9.4 release), and pictures were drawn using R (3.6.3 version). For ELISpot assays, the statistical analysis and graphical presentations were performed using GraphPad Software 8.01 (GraphPad Software Inc, La Jolla, CA, USA). Paired data were analyzed by Wilcoxon signed rank test. Statistical differences between groups were assessed by the non-parametric Mann–Whitney U test. Pearson’s correlation was used for the evaluation of bivariate associations.

## Figures and Tables

**Figure 1 ijms-24-13728-f001:**
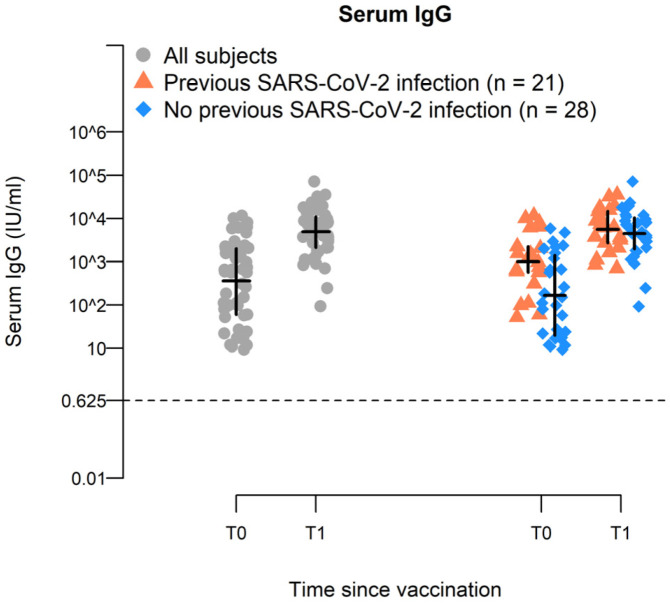
Humoral response to BNT162b2 vaccination in elderly. Distribution of serum IgG at T0 and T1 in all the recruited individuals and according to their exposure to SARS-CoV-2 infection after primary vaccination cycle. In each group, the horizontal line represents the sample median, while the vertical line represents the interquartile range. T0 = pre-booster; T1 = post-booster. Statistical analyses were reported in Table 2.

**Figure 2 ijms-24-13728-f002:**
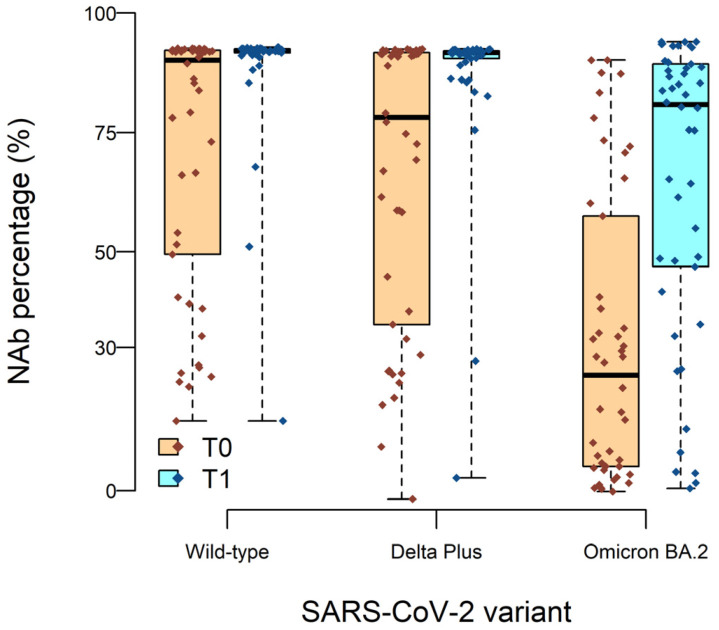
BNT162b2 booster in elderly elicits strong neutralizing antibodies activity against Wild-type and Delta Plus variants but weaker against Omicron BA.2. Nab percentage distribution of SARS-CoV-2 variants at T0 (orange) and T1 (light blue) according to SARS-CoV-2 variants.

**Figure 3 ijms-24-13728-f003:**
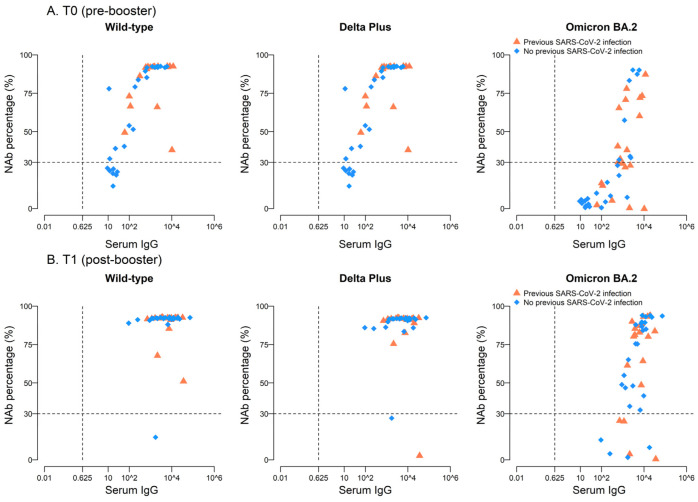
Neutralizing activity against Omicron BA.2 variant positivity correlates with IgG titer before and after the booster dose of BNT162b2. Scatter plot for Nab percentage and serum IgG at T0 (**A**) and at T1 (**B**) in all the recruited individuals and according to their exposure to SARS-CoV-2 infection after primary vaccination cycle. Dashed lines represent the reference standards values.

**Figure 4 ijms-24-13728-f004:**
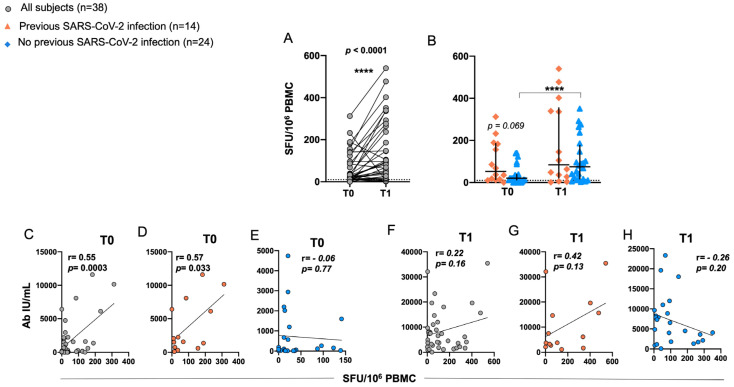
Cellular response to BNT162b2 vaccination in elderly. PBMCs collected at T0 and at T1 were stimulated for 24h with RBD-15 mer overlapping peptides. Each data point represents the normalized mean spot count from duplicate wells for one study participant, after subtraction of the non-stimulated control in the overall sample (**A**) and according to previous SARS-CoV-2 infection (**B**) Results were given as IFN-γ spot-forming units (SFU)/10^6^ PBMC. The positive cut-off was set at 10 IFN-γ SFU/10^6^ PBMC. Correlations between IgG levels and the cumulative SFU responses at T0 and T1 in overall population ((**C**,**F**), respectively), and according to prior SARS-CoV-2 infection ((**D**,**E**,**G**,**H**), respectively) as assessed by Pearson correlation. r, correlation coefficient. **** *p* < 0.00001.

**Table 1 ijms-24-13728-t001:** Demographic and clinical features of the study participants.

Variable	All Subjects (n = 49)
Age, years	84.8 ± 10.6
Sex, Female	37 (75.5%)
Number of comorbidities	1.3 ± 0.7
Comorbidities -Dementia -Cardiovascular disease (CVD) -Diabetes -COPD -Autoimmune disease	27 (55.1%) 20 (40.8%) 8 (16.3%) 7 (14.3%) 3 (6.1%)
Previous SARS-CoV-2 infection at T0	21 (42.9%)
Previous flu vaccination at T0	7 (14.3%)

In the table: mean ± standard deviation for continuous variables, and counts and (%) for categorical variables T0: before the COVID-19 vaccine booster dose. COPD: chronic obstructive pulmonary disease.

**Table 2 ijms-24-13728-t002:** Time trends for serum IgG antibodies between T0 and T1 in the overall sample and according to previous SARS-CoV-2 infection.

Time	All Subjects (n = 49)		No Previous SARS-CoV-2 (n = 28)		Previous SARS-CoV-2 (n = 21)		*p*-Value ^3^	*p*-Value ^4^
	Mean ^1^	Δ(IC 95%) ^2^	Mean ^1^	Δ(IC 95%) ^2^	Mean ^1^	Δ(IC 95%) ^2^		
T0	361	-	166	-	1011	-	0.91	<0.0001
T1	4910	4549.2 (2778; 6320.3)	4470	4304 (1877.4; 6730.6)	5563	4552.8 (707; 8398.5)		0.64

^1^: Geometric mean concentration. ^2^: change in geometric mean concentration, modelled through a log-linear regression model for repeated measures, with unstructured variance-covariance matrix. ^3^: *p*-value testing homogeneity of trends between subjects with and without previous SARS-CoV-2 infection. ^4^: *p*-value testing homogeneity of geometric mean values between subjects with and without SARS-CoV-2 infection, at each time.

**Table 3 ijms-24-13728-t003:** Correlation between serum IgG * and VOC-specific Nab positivity at T0 and T1, in the overall sample and according to previous SARS-CoV-2 infection.

	Wild-Type	Delta Plus	Omicron BA.2
All patients (n = 46), time			
T0	**0.79**	**0.72**	**0.72**
T1	0.06	−0.05	**0.53**
No previous SARS-CoV-2 (n = 26), time			
T0	**0.89**	**0.87**	**0.80**
T1	0.17	0.23	**0.64**
Previous SARS-CoV-2 (n = 20), time			
T0	0.24	0.16	**0.52**
T1	−0.21	−0.34	0.33

*: log-linear concentration. Pearson correlation coefficient. Bold: Pearson correlation coefficient different from zero (*p*-value < 0.05).

**Table 4 ijms-24-13728-t004:** Association between demographic and clinical characteristics and vaccine booster dose response (T1), assessed as serum IgG and neutralizing antibodies against Omicron BA.2 variant.

Variable	N	Serum IgG		Nab%, Omicron BA.2	
		Geometric Mean (95% CI)	*p*-Value ^	Mean (95% CI)	*p*-Value ^
Age *	49	−0.01 (−0.35; 0.33)	0.58	0.9 (−7.6; 9.4)	0.84
Time since T0 **	49	0.009 (−0.17; 0.18)	0.91	4.3 (0.2; 8.3)	0.04
Sex					
men	12	5465 (2682; 11,139)	0.73	61.6 (44.4; 78.8)	0.65
women	37	5092 (3380; 7670)		63.9 (53.9; 73.9)	
Diabetes					
Yes	8	7231 (3016; 17,341)	0.34	71.7 (50.4; 93.0)	0.50
No	41	5739 (3566; 9236)		67.7 (56.0; 79.3)	
COPD					
Yes	7	11,583 (4698; 28,553)	0.04	57.3 (34.7; 79.9)	0.50
No	42	7021 (4318; 11,416)		61.9 (49.6; 74.1)	
CVD					
Yes	20	4548 (2555; 8097)	0.71	64.9 (50.1; 79.7)	0.98
No	29	4853 (3388; 6952)		65.0 (55.9; 74.1)	
Dementia					
Yes	27	3332 (2128; 5218)	0.01	63.1 (51.2; 75.1)	0.65
No	22	5133 (3677; 7167)		65.2 (56.4; 74.0)	
Autoimmune Disease					
Yes	3	13,210 (3235; 53,945)	0.16	51.4 (16.9; 86.0)	0.43
No	46	7799 (3777; 16,104)		58.7 (40.9; 76.5)	
Flu Vaccination between T0 and T1					
Yes	21	3504 (2075; 5919)	0.10	60.6 (47.0; 74.2)	0.41
No	28	4708 (3329; 6659)		64.4 (55.5; 73.3)	

*: beta-coefficient (with 95% CI) for 10 years increase in age. **: beta-coefficient (with 95% CI) for 10 days increase in time since T0. Geometric mean: estimated from univariate log-linear regression models, adjusting for previous SARS-CoV-2 infection. Mean: estimated from univariate linear regression models, adjusting for previous SARS-CoV-2 infection. ^: Wald chi-square *p*-value from univariate regression models, adjusting for previous SARS-CoV-2 infection. CVD: Cardiovascular disease. COPD: chronic obstructive pulmonary disease.

## Data Availability

The data presented in this study are available on request from the corresponding author. The data are not publicly available due to privacy restrictions.

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
