# Peer review of "Humoral and Cellular Immune Response Elicited by the BNT162b2 COVID-19 Vaccine Booster in Elderly"

_ijms, 2023, doi:10.3390/ijms241813728_

Round 1

Reviewer 1 Report

The manuscript studied serum antibody and T cell responses among long-term care facility (LTCF) residents before and after BNT162b2 booster (third dose). Neutralizing antibody titers and T cell activities were analyzed. Data analysis was careful and stringent. Here are some suggestions:

1. Omicron BA.2 is no longer the current circulating strain. It might be helpful to add neutralization assays for the recent circulating strains.

2. Some interesting correlation was observed between T cell responses and anti-Spike antibody titers. What does this mean? The significance of these correlations should at least be further discussed in the discussion section.

The manuscript is well-written. There are some minor grammar mistakes or typos that need to be fixed, but those are trivial.

Author Response

The manuscript studied serum antibody and T cell responses among long-term care facility (LTCF) residents before and after BNT162b2 booster (third dose). Neutralizing antibody titers and T cell activities were analyzed. Data analysis was careful and stringent. Here are some suggestions:

  1. Omicron BA.2 is no longer the current circulating strain. It might be helpful to add neutralization assays for the recent circulating strains.

We thank the reviewer for the comment. In this work we decided to test the Delta Plus and more importantly Omicron BA.2 VOCs because they were the most circulating viral strains at the time of sample collection. For some subjects, serum samples have been fully used for this study and thus assessing the titer of neutralizing antibodies against the present variants would be performed for a reduced number of subjects. Moreover, most of the subjects recruited in this study, have been subsequently vaccinated with a fourth dose of BNT162b2. Thus, when completed the vaccination protocol, it will certainly be interesting to evaluate the neutralizing antibody titers against the present circulating Omicron subvariants as XBB generated by the combination of two strains of Omicron subvariant BA.2 and comparing the protection between pre- and post- fourth dose booster.

  1. Some interesting correlation was observed between T cell responses and anti-Spike antibody titers. What does this mean? The significance of these correlations should at least be further discussed in the discussion section.

We thank the reviewer for the comment.

The correlation between T cells responses and anti-spike antibody titers was previously observed by others particularly in naturally infected pre-exposed individuals (Österdahl MF, Christakou E, Hart D, Harris F, Shahrabi Y, Pollock E, Wadud M, Spector TD, Brown MA, Seow J, Malim MH, Steves CJ, Doores KJ, Duncan EL, Tree T. Concordance of B- and T-cell responses to SARS-CoV-2 infection, irrespective of symptoms suggestive of COVID-19. J Med Virol. 2022 Nov;94(11):5217-5224. doi: 10.1002/jmv.28016. Epub 2022 Jul 30.)

Higher anti-spike Ab titers and correspondingly high specific T cells most likely correlate to the increased number of memory T cells generated by vaccination/natural infection and thus to the consequent increased primed T cell-dependent stimulation of antibody producing B cells.

A new reference (new reference 40) was added, and the comment was added in the Discussion section (lines 348-353)

Reviewer 2 Report

In this manuscript, the authors studied the immune responses of elderly residents in long-term facilities against COVID-19 before and after receiving the BNT162b2 mRNA vaccine booster and explored the potential associations with sex, age, and comorbidities.  Some minor comments are provided for authors to address:

1. The authors evaluated the neutralizing titers against wild-type, delta plus and Omicron BA2. Could authors also evaluate the titers against the latest Omicron variant (e.g., one of the XBB subvariants) to make it more relevant to the current situation?

2. In Table 4, the numbers of men and women are listed as 37 and 12, respectively, but Line 82 and Table 1 indicate that there were 37 female subjects. please verify the number and correct it accordingly.

3. A relatively similar study published by Hansen et al 2023 also evaluated the immune responses of old adults after receiving BNT162b2  (https://www.ncbi.nlm.nih.gov/pmc/articles/PMC9830931/). Could authors in the Discussion section compare your and their studies?

Author Response

In this manuscript, the authors studied the immune responses of elderly residents in long-term facilities against COVID-19 before and after receiving the BNT162b2 mRNA vaccine booster and explored the potential associations with sex, age, and comorbidities.  Some minor comments are provided for authors to address:

  1. The authors evaluated the neutralizing titers against wild-type, delta plus and Omicron BA2. Could authors also evaluate the titers against the latest Omicron variant (e.g., one of the XBB subvariants) to make it more relevant to the current situation?

We thank the reviewer for the comment. In this work we decided to test the Delta Plus and more importantly Omicron BA.2 VOCs because they were the most circulating viral strains at the time of sample collection. For some subjects, serum samples have been fully used for this study and thus assessing the titer of neutralizing antibodies against the present variants would be performed for a reduced number of subjects. Moreover, most of the subjects recruited in this study, have been subsequently vaccinated with a fourth dose of BNT162b2. Thus, when completed the vaccination protocol, it will certainly be interesting to evaluate the neutralizing antibody titers against the present circulating Omicron subvariants as XBB generated by the combination of two strains of Omicron subvariant BA.2 and comparing the protection between pre- and post- fourth dose booster.

  1. In Table 4, the numbers of men and women are listed as 37 and 12, respectively, but Line 82 and Table 1 indicate that there were 37 female subjects. please verify the number and correct it accordingly.

We apologies for the mistake. We have modified the number of the female in Table 4.

  1. A relatively similar study published by Hansen et al 2023 also evaluated the immune responses of old adults after receiving BNT162b2(https://www.ncbi.nlm.nih.gov/pmc/articles/PMC9830931/). Could authors in the Discussion section compare your and their studies?

We thank the reviewer for the suggestion. Hansen et al have evaluated both humoral and cellular immune response in naturally infected older people and in uninfected older people nine months after the first dose of BNT162b2 vaccine. Although this latter kinetic point is comparable to our T0 samples, it must be noted that our comparison was made between vaccinated people with or without previous natural exposure to viral infection.

They observed that T-cell responses persisted up to 12-months only in naturally infected older patients but waned in previously unexposed subjects. Accordingly, in our study T cell response was low at T0 in vaccinated older people. Importantly, and similarly to our results, they observed that pre-infected older adults had more robust and durable antibody responses compared to unexposed individuals. Cumulatively, these results emphasize that natural infection induces more robust and durable immune responses in the elderly, which is not achieved by two doses of BNT162b2 vaccine.

As asked by the reviewer, we added the new reference (new references 39) and related comments in the Discussion section, lines 336-345.

New reference 39: Hansen L, Brokstad KA, Bansal A, Zhou F, Bredholt G, Onyango TB, Sandnes HH, Elyanow R, Madsen A, Trieu MC, Sævik M, Søyland H, Olofsson JS, Vahokoski J, Ertesvåg NU, Fjelltveit EB, Shafiani S, Tøndel C, Chapman H, Kaplan I, Mohn KGI, Langeland N, Cox RJ. Durable immune responses after BNT162b2 vaccination in home-dwelling old adults. Vaccine X. 2023 Apr;13:100262. doi: 10.1016/j.jvacx.2023.100262. Epub 2023 Jan 10. PMID: 36643855

Round 2

Reviewer 1 Report

Thank you for answering my questions/comments.

Some minor edits may be needed.